


# On the skill of raw and postprocessed ensemble seasonal meteorological forecasts in Denmark

Diana Lucatero[1], Henrik Madsen[2], Jens C. Refsgaard[3], Jacob Kidmose[3], Karsten H. Jensen[1]

[1]Department of Geosciences and Natural Resource Management, University of Copenhagen, Copenhagen, Denmark

[2] DHI, Hørsholm, Denmark

[3] Geological Survey of Denmark and Greenland (GEUS), Copenhagen, Denmark

*Correspondence to:* Diana Lucatero (diana.lucatero@ign.ku.dk)

**Abstract.** This study analyzes the quality of the raw and postprocessed seasonal forecasts of the European Center of Medium Weather Forecasts (ECMWF) System 4. The focus is given to Denmark located in a region where seasonal

forecasting is of special difficulty. The extent to which there are improvements after postprocessing is investigated. We make use of two techniques, namely, linear scaling/delta change (LS) and quantile mapping (QM) to daily bias correct seasonal ensemble predictions of hydrological relevant variables such as precipitation (P), temperature (T) and reference evapotranspiration ($E_{T0}$). Qualities of importance in this study are the reduction of bias and the improvement in accuracy and sharpness over ensemble climatology. Statistical consistency and its improvement is also examined. Raw forecasts exhibit

biases in the mean that have a spatio-temporal variability more pronounced for P and T. This variability is more stable for $E_{T0}$ with a consistent positive bias. Accuracy is higher than ensemble climatology for some months at the first month lead time only and, in general, ECMWF System 4 forecasts tend to be sharper. $E_{T0}$ also exhibits an underdispersion issue, i.e., forecasts are narrower than their true uncertainty level. After correction, reductions in the mean are seen. This however, is not enough to ensure an overall higher level of skill in terms of accuracy although modest improvements are seen for T and

$E_{T0}$, mainly at the first month lead time. QM is better suited to improve statistical consistency of forecasts that exhibit dispersion issues, i.e., when forecasts are consistently overconfident. Furthermore, it also enhances the accuracy of the monthly number of dry days to a higher extent than LS. Caution is advised when applying a multiplicative factor to bias correct variables such as P. It may overestimate the ability that LS has in improving sharpness when a positive bias in the mean exists.

## 1 Introduction

Seasonal forecasting has gained increasing attention during the last three decades due to high societal impacts of extreme meteorological events that affect a plethora of weather-related sectors such as agriculture, environment, health, transport and energy, and tourism (Dessai and Soares, 2013). Information of weather-related hazards months ahead are important for protection against extremes for these sectors.

General Circulation Models (GCM) have become the state-of-the-art technology for issuing meteorological forecasts at different time scales. GCM-based seasonal forecasting is possible due to signals that can be extracted from slowly changing systems such as the ocean, and to a lesser extent, land, that then translates into a signal in the atmospheric patterns (Weisheimer and Palmer, 2014; Doblas-Reyes et al., 2013). El Nino Southern Oscillation, ENSO, is the strongest of these signals, and its influence on seasonal forecasting is higher near the tropics (Weisheimer and Palmer, 2014).

Seasonal ensemble forecasts have been operational in Europe since the late 1990s provided by the European Center for Medium Range Forecast (ECMWF) (Molteni et al., 2011) and in the U.S. since August 2004 provided by the National Center of Environmental Prediction (Saha et al., 2013). Other examples of operational seasonal forecasts include the ones





generated by the Met Office in UK (Maclachlan et al., 2015), the Australian Bureau of Meteorology (Hudson et al., 2013), the Beijing Climate Center (Liu et al., 2015) and the Hydrometeorological Center of Russia (Tolstykh et al., 2014).

ECMWF is a leading center for weather and climate predictions and its seasonal forecasting system is often regarded as the best (Weisheimer and Palmer, 2014). Research on the quality of the atmospheric forecasts has been done for different system

versions (Molteni et al., 2011; Weisheimer et al., 2011). The system has also been compared to other GCM (Kim et al., 2012a, 2012b; Doblas-Reyes et al., 2013) or statistical (van Oldenborgh et al., 2005) seasonal forecasting systems.

Despite of the efforts mentioned above and the documented improvements on forecasting skill of meteorological parameters, specially over the tropics (Molteni et al., 2011), several issues still remain. The main one, and specific to forecasting in Europe and North America is that the signal of the main driver of seasonal predictability, the ENSO, has been found to be

weak or non-existent (Molteni et al., 2011; Saha et al., 2013) in these regions leading to poor skill of atmospheric variables such as precipitation. For example, Weisheimer and Palmer, (2014) studied the reliability (consistency between the forecasted probabilities and their observed frequencies) and ranked forecasts using five categories from 'dangerous' (1) to 'perfect' (5) for two regimes of precipitation (wet/dry) and temperature (cold/warm). For the North European region, they found dry (wet) forecasts during summer, started in May to be 'dangerous' ('marginally useful') and dry (wet) forecasts

during winter (started in November) to be 'not useful' ('marginally useful'). For temperature, results were less variable among the different categories with winter cold/warm and summer warm forecasts found to be 'marginally useful', and summer cold temperatures forecasts in the category (5) for 'perfect'. Moreover, Molteni et al., (2011) found weak anomaly correlations of precipitation and temperature during the summer for most of the regions located in North Europe.

Due to the issues stated above, the need for postprocessing the raw forecasts in the hope of improvements has gained

importance in the scientific literature. A plethora of methods for statistical postprocessing exist for a range of temporal scales. These methods consist on transfer functions, computed on the basis of reforecasts, or past records of forecast-observation pairs (Hamill et al., 2004) whose goal is to match forecast values with observed ones. The choice of postprocessing method is determined by the availability of reforecast data and the application at hand. Although in principle any method could be used for seasonal forecasts, this temporal scale represents a special difficulty due to the fact that initial

condition skill is mostly gone and there is little detectable signal behind a large amount of chaotic error.

In particular, for the postprocessing of ECMWF System 4 seasonal forecasts, a number of studies have been carried out: Crochemore et al., (2016); Peng et al., (2014); Trambauer et al., (2015) and Wetterhall et al., (2015). The most used methods are linear scaling and quantile mapping, although Peng et al. (2014) used a Bayesian Merging technique. In general, the aforementioned studies are successful in improving the values of the forecast qualities that they considered important. For

example, Wetterhall et al., (2015) reported higher skill of forecasts of the frequency and duration of dry spells once an empirical quantile mapping has been applied to daily values of precipitation. Crochemore et al., (2016) analyzed the effect different implementations of the linear scaling and quantile mapping methods had on streamflow forecasting, concluding that the empirical quantile mapping improves the statistical consistency of the precipitation forecasts for different catchments throughout France.

The aforementioned studies have been made for precipitation and/or mainly large areas. For hydrological applications seasonal forecasting skill of instantaneous values of precipitation (P), temperature (T) and reference evapotranspiration ($E_{T0}$) at the catchment scale (100 - 1,000 $km^2$) are, however, more important. Therefore, we analyze the bias, skill and statistical consistency of the ECMWF System 4 for Denmark focusing on P, T and $E_{T0}$ of relevance for seasonal streamflow forecasting at catchment scale. We make use of the two most used methods for postprocessing, namely linear scaling and

quantile mapping, applied to daily values. We focus on the skill of monthly aggregated values of gridded data throughout Denmark for both the raw and the corrected forecasts. We attempt to answer the following questions:



(1) what is the longest lead time for which an 'acceptable' forecast is achieved?

(2) is it possible to extend the acceptable forecast lead time with different postprocessing techniques?

In this study we argue that, an acceptable forecast needs to have consistency between the observed probability distribution and the predictive one, this is what we call statistical consistency throughout the paper. A statistical consistent forecast system has low (or non-existent) bias in both mean and variance. Secondly, we argue that the forecast to be used has to be better than climatology, having a higher skill both in terms of accuracy and sharpness, giving priority to the former. These characteristics for an 'acceptable forecast' follow the principle that the purpose of postprocessing is to maximize sharpness subject to statistical consistency as discussed by Gneiting, et al., (2007).

## 2 Data and Methods

### 2.1 Ensemble Prediction System and Observational Grid

Seasonal reforecast of the ECMWF System 4 for the years 1990-2013 are used in the present study. The system is comprised by 15 members (for January, March, April, June, July, September, October, December) and 51 members (for February, May, August, November) with a spatial resolution of 0.7 degrees and are run for seven months with daily output. P, T and $E_{T0}$ are the variables under study. For the computation of $E_{T0}$, we make use of the Makkink equation (Hendriks, 2010) that takes as inputs temperature and incoming short-wave solar radiation from ECMWF System 4.

Observed daily values for P, T and $E_{T0}$ from the Danish Meteorological Institute (DMI) are used (Scharling and Kern-Hansen, 2012). The spatial scale for P and T, $E_{T0}$ is 10 km and 20 km, respectively. However, we assume T and $E_{T0}$ to be equally distributed within the 20 km and set the same values of the 20 km to the 10 km grid. Then, in total there are 662 (for P) and 724 (for T and $E_{T0}$) grid points that cover the 43,000 $km^2$ area of Denmark. Moreover, P is corrected for under catch errors as explained in Stisen, et. al., (2011) and (2012). The time and spatial variations of the variables can be seen in Fig. 1. Values are monthly accumulations for P and $E_{T0}$ and monthly averages for T, averaged over the observed record (1990-2013). Danish weather is mainly driven by its proximity to the sea. There is a modest spatial P gradient from west to east which is more pronounced during autumn and winter. The driest month in terms of P is April and the wettest is October. $E_{T0}$ also shows a modest spatial variability during spring and summer with larger values in eastern Denmark.

### 2.2 Postprocessing strategy

Given the fact that both the ensemble and observed spatial resolutions differ, first the ensemble forecasts were interpolated to match the 10 km grid of observed values using an Inverse Distance Weighting (Shepard, 1968), where the values at a given point of the higher resolution grid (10 km) are computed using a weighted average of the four surrounding nodes of the lower resolution forecast grid (70 km). The weights are computed as the inverse of the Euclidean distances between the observed grid node and the forecast nodes. Forecasts are then postprocessed for each grid point, time of forecast (month), and lead time (month) for each variable separately. Moreover, the computation is done in a leave-one-out cross-validation mode (Wilks, 2011 and Mason and Baddour, 2008) such that the year that we are correcting is withdrawn from the training sample to ensure independence between training and validation data. Then, for example for precipitation, 662x12x7x24 correction models are computed.

### 2.3 Postprocessing methods

#### 2.3.1 Delta method - Linear Scaling (LS)

The linear scaling approach operates under the assumption that forecast values and observations will agree in their monthly mean once a scale factor has been applied (Teutschbein and Seibert, 2012). LS is the simplest possible postprocessing method as it only corrects for biases in the mean. The factor is commonly computed differently for P, $E_{T0}$ and T.





For P and E$_{T0}$:

$$f_{k,i}^* = \frac{\dfrac{1}{N}\sum_{i=1}^{N} y_i}{\dfrac{1}{N}\sum_{i=1}^{N} \bar{f_i}} f_{k,i} \tag{1}$$

For temperature:

$$f_{k,i}^* = f_{k,i} - \frac{1}{N}\left[\sum_{i=1}^{N} \bar{f_i} - \sum_{i=1}^{N} y_i\right] \tag{2}$$

where $f_{k,i}$ denotes ensemble member $k$ for $k=1,\dots,M$ of forecast-observation pair $i=1,\dots,N$, $M$ denotes the number

of members (15 or 51) and $N$ is the number of forecast-observation pairs, $\bar{f_i}$ denotes the ensemble mean, $y_i$ denotes the

verifying observation. Note that, as stated in Sect. 2.3., both the means of $\bar{f_i}$ and $y_i$ are computed with the sample that

withdraws forecast and observation pair $i$. Finally, $f_{k,i}^*$ represents the corrected ensemble member.

### 2.3.2 Quantile mapping (QM)

QM relies on the idea of Panofsky and Brier (1968). This method matches the quantiles of the predictive and observed

distribution functions in the following way:

$$f_{k,i}^* = G_i^{-1}\left(F_i\left(f_{k,i}\right)\right) \tag{3}$$

where $F_i$ represents the predictive cumulative distribution function (CDF) for forecast-observation pair $i$, $G_i$ represents the

observed CDF. Again, note that, as stated in Sect. 2.3., both $F_i$ and $G_i$ are computed with a sample that withdraws forecast

and observation pair $i$.

$F_i$ is calculated as an empirical distribution function trained with all ensemble members of daily values of a given month for

a given lead time and grid point. For example, for a forecast of target month June initialized in May, $F_i$ is trained using a

sample comprising 30 (days) times 23 (number of years in the reforecast minus the year to be corrected) times 51 (number of

ensemble members). The same is done for $G_i$, except that the training sample is comprised by 30 x 23 values only. $F$ and $G$

are computed as an empirical CDF. Linear interpolation is needed in order to approximate the values between the bins of $F$

and $G$. Extrapolation is then needed to map ensemble values and percentiles that are outside the training range.

### 2.4 Verification metrics

As a manner to evaluate first the raw forecasts and the improvement after postprocessing we check for four qualities: bias,

skill in regards to accuracy and sharpness, and statistical consistency.

### 2.4.1 Bias

Bias is a measure of under – overestimation of the mean of the ensemble in comparison with to the observed mean:


$$\%Bias = \left( \frac{\sum_{i=1}^{N} \bar{f}_i}{\sum_{i=1}^{N} y_i} - 1 \right) \times 100 \tag{4}$$

for P and $E_{T0}$ and:

$$Bias = \frac{1}{N} \left[ \sum_{i=1}^{N} \bar{f}_i - \sum_{i=1}^{N} y_i \right] \tag{5}$$

for T. $f_i$ and $y_i$ are the same as in Eq. (1).

If the bias is negative, the forecasting system then exhibits a systematic underprediction. Conversely, if the amount is positive the system shows an average overprediction. Values closer to 0 are of course desirable.

### 2.4.2 Skill

The skill of a forecasting system is the improvement, on average, that the system has with respect to a reference system that could be used instead, for example climatology for seasonal forecasts or persistence for short-range forecasts. The skill score
is computed in the following manner

$$Skill = \frac{Score_{sys} - Score_{ref}}{Score_{per} - Score_{ref}} \tag{6}$$

where $Score_{sys}$, $Score_{ref}$ and $Score_{per}$ are the score value of the system to be evaluated, the reference system and the value of a perfect system, respectively. The range of the skill is from $-\infty$ to 1 and values closer to 1 are preferred. In this paper we calculate the skill with respect to accuracy and sharpness. We compute the continuous rank probability score
(CRPS) (Hersbach, 2000), as a general measure of the accuracy of the forecast. The computation of the score is as follows:

$$CRPS = \frac{1}{N} \sum_{i=1}^{N} \int_{-\infty}^{\infty} [P_i(x) - H(x - y_i)]^2 dx, \tag{7}$$

where $P_i(x)$ is the CDF of the ensemble forecast for pair $i$ and $H(x - y_i)$ the Heaviside function that takes the value 1 when $x > y_i$ and 0 otherwise, $y_i$ and $N$ are, as in Eq. (1), the verifying observation for forecast-observation pair $i$, and the number of forecast-observation pairs, respectively. We made use of the EnsCrps function of the R package SpecsVerification
(Siegert, 2015) developed in R version 0.4-1. For the skill with respect to sharpness we use the average along $i = 1, \dots, N$ of the differences between the 25% and the 75% percentiles of each of the ensemble CDFs, $P_i$.

In Eq. (6) our reference forecast is ensemble climatology (1990-2013) where the year to be evaluated is withdrawn from the sample. Both the accuracy and sharpness score for a perfect system cf. Eq. (6), $Score_{per}$ is equal to zero so the skill score can be then simplified as:

$$Skill = 1 - \frac{Score_{sys}}{Score_{ref}} \tag{8}$$

which, once multiplied by 100, can be seen as the percentage of improvement (if positive) or worsening (if negative) over the reference forecast. Throughout the paper, the skill related to accuracy will be denoted as CRPSS whereas the skill due to sharpness will be denoted as SS.



Furthermore, in order to define the statistical significance of the differences between the skill of ensemble climatology and ECMWF System 4 forecasts, as well as the postprocessed predictions, a Wilcoxon-Mann-Whitney test (WMW-test; see Hollander et al, 2014) was carried out. The WMW test, unlike the most common t-test, makes no assumptions about the underlying distributions of the samples.

### 2.4.3 Statistical consistency

We use the Probability Integral Transform (PIT) diagram for a depiction of the statistical consistency of the system. The PIT diagram is the CDF of the $z_i$'s defined as $z_i = P_i(X \leq y_i)$. Therefore, $z_i$ is the value that the verifying observation $y_i$ attains within the ensemble CDF, $P_i$. The diagram represents an easy check of the biases in the mean and dispersion of the forecasting system. For a forecasting system to be consistent, meaning that the observations can be seen as a draw of the

forecast CDF, a quality that a forecasting system should aim for, the CDF of $z_i$ should be close to the CDF of a uniform distribution on the [0,1] range. Deviations from the 1:1 diagonal represent bias issues in the ensemble mean and spread as explained in appendix A and shown in Fig. A12. Similar to (Laio and Tamea, 2007), we make use of the Kolmogorov bands to have a proper graphical statistical test for uniformity.

### 2.5 Accuracy of maximum monthly daily precipitation and number of dry days

For applications such as flooding and forecasting of low flows and droughts, water managers might be interested not only in the skill of monthly accumulated precipitation but also in the skill of other precipitation quantities. We will use a rather simple approach for checking the deficiencies of the raw forecasts and whether the postprocessing methods improve these deficiencies by analyzing the improvement in the prediction of also monthly maximum daily precipitation and number of dry days in a given month. For the purpose of this study, a dry day is defined as the day with observed zero-precipitation, and the

comparison with the ensembles is made on a daily basis.

## 3 Results

### 3.1 Analysis of raw forecasts

The first row in Fig. 2 depicts the ECMWF System 4 forecast and an ensemble climatological forecast for August accumulated P and $E_{T0}$ and averaged T for one grid located in west-central Denmark for the first month lead time. The values

for different forecast qualities for that grid point are also included. For a forecasting system to be useful, it has to be at least, better than a climatological forecast. For the given example here, we show in the background the reference forecast that is wider than the ECMWF System 4 forecast. This is an example of a month where we have a slightly better skill than the ensemble climatological forecast for the three variables in question. For example, raw T predictions from ECMWF System 4 improve, on average, on the reference forecast by 20% in terms of accuracy. This level of skill is attained due to the sharper

forecasts that exhibit a low bias (-0.23 deg C). On the other hand, sharpness is only a desirable property when biases are low. This is illustrated for P forecasts attaining a high skill due to sharpness (0.43) but at the expense of a low skill due to accuracy (0.01) that is caused by the high negative bias (-14.12%) where, for example for 1992 and 2010, the verifying observation lies outside the ensemble range contributing negatively to the CRPS in Eq. (7).

### 3.1.1 Bias

In an effort to summarize the results, a spatial average of the bias throughout Denmark was computed. Figure 3 shows the spatial average bias of P, T and $E_{T0}$ of the raw forecasts. Y-axis represents the target month, for example April, and X-axis represents the forecast lead time, lead time 5 is the forecast for April initiated in December. As we can see from Fig. 3, bias depends on the target month and, to a lesser degree, on lead time. For P, the lowest bias can be found throughout autumn to beginning of winter, followed by a general underestimation of P that is at its highest for June. April shows an overestimation

that might be due to the 'drizzle effect' in a month where dry days are more common. The drizzle effect issue is a very well-





known problem of GCM, and is related to the generation of small precipitation amounts, usually around 1.0-1.5 mm/day where observed precipitation is not present (Wetterhall et al., 2015).

T bias averaged over Denmark has a range that lies within [-2,2] degrees Celsius. The bias switches from positive to negative when temperatures start to increase in March and from negative to positive bias when temperatures start decreasing in
August. This indicates that the forecast of T has a smaller annual amplitude than observed. Lowest biases are encountered during January and February with a bias of 0.5 deg Celsius, and it is higher during late spring and summer, with a negative bias of almost 2 deg Celsius. Finally, the bias range for $E_{T0}$ is smaller than for P taking values within [0-25%] on average over Denmark. In general, there is a positive bias, which is at its highest during February.

However, averaging does not tell the whole story. We are also interested in the spatial variation of biases over Denmark.
Figure 4 shows the spatial distribution of bias for the first month lead time and its evolution during summer. In general, there is an underestimation of P throughout Denmark, much more pronounced during June. Nevertheless there also exists a positive bias in central Jutland and on the urban area of Copenhagen reaching a value around 10-20%. The positive bias area grows in July occupying most of Jutland and North Zealand.

Other seasons were also mapped and shown in the supplements as Fig. S1 to Fig. S3. During autumn and winter there is also
a general negative bias that is more pronounced in central Jutland, reaching values of -30%. Nevertheless, an overestimation exists in eastern Denmark for those seasons. For winter, this overestimation is present in the sea grid points. Finally, during spring the spatial variability changes. For example, most grid points exhibit a positive bias during April, except for the southeast region of Denmark that has a small negative bias between 0.0 to 5.0%. During May a tendency of overforecasting is present in central Jutland.

The spatial distribution of T bias during autumn and winter (Fig. S2 and Fig. S3, respectively) follow a similar pattern with a general positive bias reaching its highest values in the southeast region (from 1.5 to 2.0 deg C). A negative bias is seen during spring (Fig. S1) and late summer across Denmark (Fig. 4). In June a positive bias [0-2 deg C] is present in a large area of the Jutland peninsula (Fig. 4). Finally, the spatial variation of bias of $E_{T0}$ is less pronounced and, in general, positive. Nonetheless, exceptions exist. There is a negative bias in small regions located in the coastal areas or sea grid points, that
ranges from -10 to 0%.

The results presented above are specifically for lead time 1, i.e., forecasts of accumulated P and $E_{T0}$ and average T for August initialized on August 1$^{st}$. The spatial variation of bias for other lead times was also mapped (not shown) and analyzed. In general, similar spatial patterns were found for all three variables, being the same along the target months regardless of lead time.

**3.1.2 Skill**

Figure 5 summarizes the results for skill in the following manner. First, the values presented are, like in Fig. 3, the spatial average of skill for entire Denmark. Secondly, we compare the skill in terms of accuracy and sharpness and visualize it in a format that can give us both skill comparisons at the same time. Accuracy and sharpness skill scores were binned into four categories with the first one being the case where, on average, ECMWF System 4 is scoring worse than a forecast based on
ensemble climatology. The remaining bins represent the different levels of skill where System 4 scores better than the reference forecast.

The bins for skill with respect to accuracy are represented with different colors, whereas the bins for skill with respect to sharpness are represented with different shapes/sizes, the bigger the size the wider the ensemble from System 4 is with respect to ensemble climatology, on average. The squares represent the case where ECMWF System 4 has a negative skill





score, indicating that climatological forecasts are sharper than those of System 4. As for bias, skill appears to be dependent on the target month and, to a lesser degree, on lead time.

For P, and looking at the first month lead time, ECMWF System 4 skill in accuracy mildly beats that of climatology for February, March, April, July, August, November and December, with a CRPSS of 0.15 at most. In general, skill in accuracy decreases for lead time 2 onwards. April stands out for having a slightly positive skill in accuracy for almost all lead times, but comes at the expense of having a wider spread than ensemble climatology. For T, for the first month lead time, a positive skill exists in terms of accuracy for almost all months, except for late spring. Skill in terms of accuracy also decreases with lead time, but February stands out to have a mild skill for almost all lead times. Forecasts are also in general sharper than ensemble climatology, with the exception of January and March at longer lead times. $E_{T0}$ appears to have skill only for late summer and beginning of autumn in terms of accuracy. This may be explained by the fact that forecasts are sharper than climatology, indicating that there could be an underdispersion issue.

Figure 6 shows the skill in accuracy (CRPSS) for monthly values and for lead time of one month mapped across Denmark and its monthly evolution during the summer. The rest of the seasons are also mapped and analyzed. These are included in the supplement as Fig. S4 to Fig. S6. Higher skill is observed for T, for which ECMWF System 4 improves the ensemble climatological forecast with up to 50%. P and $E_{T0}$ have lower skill in comparison to T, reaching a value of 0.3 for specific months and regions in Denmark. The spatial variation of skill for P seems scattered across Denmark and also through the year. Some notable exceptions are the higher skill in accuracy that ECMWF System 4 has in western Jutland during November and December and the low skill attained in October across Denmark (Fig. S5 and Fig. S6). The spatial variation of skill in accuracy of monthly averaged T seems more pronounced during autumn and spring, with northern Denmark attaining the highest values of skill for these seasons. For the remaining seasons, skill over climatology is present across the country, except for late spring where eastern Denmark has the largest negative skill. Finally, the spatial variation of skill of $E_{T0}$ is more pronounced for the months April to November with both positive and negative skill. In general, in this period eastern Denmark attains positive, although mild for some months, values of skill, except for November.

In general, the areas with highest biases shown in Fig. 4 are associated with the lowest skill scores. For instance, for October P in southern central Jutland the negative bias reaches values around 30-40 % (Fig. S2), leading to values of CRPS almost 60% smaller than that of ensemble climatology (Fig. S5). The opposite also holds, areas where biases are lower, tend to have the highest benefits over ensemble climatology, i.e., March P across Denmark or November P in western Jutland.

Skill related to accuracy was also mapped for lead times 2-7 months (not shown). In general, regions having a statistically significant positive skill score for lead time 1 month vanish, except for some smaller regions where a slight positive skill, between 0.0 and 0.1 is found, i.e. April P forecast initiated in February (lead time 3 months), which contributes to the mild positive skill at longer lead times as seen in Fig 5.

The skill related to sharpness was also mapped for all target months and lead times (not shown). In general, forecasts are sharper than ensemble climatology as seen in Fig. 5 across Denmark for all three variables under study. This situation persists, in general, along all lead times, except for April and October P and January, March and November T, as shown also in Fig. 5. For these months, the lack of sharpness is present thorough Denmark. Nevertheless, for April P, the region with the lack of sharpness is located in southern Denmark, along all lead times.

### 3.1.3 Statistical consistency of monthly aggregated P, T and $E_{T0}$

We follow Fig. A12 in Appendix A for the interpretation of the PIT diagram. The first row in Fig. 7 shows the PIT diagram for raw ECMWF System 4 summer forecasts for lead time 1 month. The remaining seasons can be seen in Fig. S7 to Fig. S9 in the supplementary material. Raw P forecasts exhibit an underprediction of the mean for winter, summer and autumn. This bias is somehow reduced for spring, except for April, where the system exhibits a positive bias. Raw T predictions of winter



and autumn, in addition to June, exhibit an overprediction, which is lowest for January. Spring and summer T exhibit a underprediction which is highest for April (Fig. S7 and Fig. 7).

Finally, raw forecasts of $E_{T0}$ during all seasons, exhibit an underdispersion, i.e. the majority of the verifying observations lie on the tails or outside the ensemble range. This is a consequence of a forecast with insufficient spread. Note, however that the underdispersivity is not present in selected months: June, November and February.

Issues with bias in the ensemble spread of P and T are not remarkably clear from the visualization of the PIT diagrams in Fig. 7. This situation is perhaps explained by the fact that we are analyzing the statistical consistency of monthly aggregated values which smoothes out extremes. The statistical consistency at longer lead times for all variables (2-7 months, not shown), depends, similarly to bias, on the target month regardless of lead time.

**3.2 Analysis of postprocessed forecasts**

The second and third columns column in Fig. 2 show the corrected forecast and the bias and skill scores after postprocessing using the LS and the QM method, respectively. The results represent a particular grid point and forecast of August initialized August 1. After postprocessing, the reduction of bias is evident for the three variables under study. Nevertheless, and contrary to what one should expect, this reduction of bias does not necessarily translate into an increase of skill in accuracy, at least for P and T and for this particular month and grid point. The quantification of the reduction/increase of accuracy after postprocessing for the whole Denmark, through the year and for different lead times is discussed in the Sect. 3.2.2 below.

**3.2.1 Bias**

Any postprocessing technique used should be able to at least remove biases in the mean. This is accomplished using both techniques. Figure 8 shows the bias of P, T and $E_{T0}$ and its evolution through the year for lead time 1 month. Bias is shown for four locations scattered around Denmark. Figure 8 shows that the yearly variability of the bias is collapsed to almost 0%, although for P and winter $E_{T0}$, the LS method seems to be doing a slightly better job at removing the bias than the QM. This comes as no surprise as the LS method forces this bias to be zero.

**3.2.2 Skill**

In order to be more quantitative in terms of the improvement over the raw forecast, we counted the number of grid points for which the skill score was positive and the number of grid cells for which the skill score was negative. Furthermore, the scores are only considered positive or negative if the differences in the distribution of the skill between ensemble climatology and ECMWF System 4 forecasts are statistical significant at the 0.05 level using the WMW test. Consequently, we introduced a third category for which there is no statistical significant difference in skill between climatology and ECMWF System 4 forecasts.

Figure 9 shows the percentage of grid cells with a statistical significant positive skill due to accuracy, Eq. (8), for the raw forecasts (first raw) and the postprocessed forecasts (LS, second row; QM third row). All target months and lead times are included. If a postprocessing method is successful in increasing the regions with positive skill scores, then the box for that particular target month/lead time is bluer in comparison to the raw forecasts. For P, there is no obvious increase in skill due to accuracy, except, perhaps, February and July forecasts for the first month lead time. There are, however, instances for which the percentage of positive skill grid points decreases. The most obvious cases are March and November (1st month lead time) with a reduction of almost half, i.e., from 13.6% (raw) to 5.7% (LS) for March. On the contrary, T and $E_{T0}$ exhibit a greater improvement, at least on the first month lead time. For instance, the percentage of grid points with positive skill increases from 4.5% to 50% for April T (LS) and from 30% to 100% (LS) for July T. The biggest improvement for $E_{T0}$ appears in June (first month lead time), reaching 90% of positive grid cells after postprocessing (LS).





In addition, the negative and equal categories were also plotted and included in the supplement as Fig. S10 and Fig. S11. After postprocessing, there are instances where a considerable amount of grid cells move from a statistical significant negative skill score to the third category (no significant differences between ensemble climatology and ECMWF System 4 score distributions, Fig. S11). This is true for T and $E_{T0}$ at longer lead times. One of the obvious examples is February $E_{T0}$ at

lead time 6 (forecast initiated in September), the percentage of grid points with negative skill scores decrease from 80.5% to 4.3% after postprocessing (Fig. S10). On the other hand the percentage of grid points with no significant differences in skill increase from 20% to 95.7% after postprocessing (Fig. S11) for this particular example.

To further illustrate the above situation, Fig. 10 shows the spatial distribution of skill due to accuracy encapsulated in box-plots that represent the 662/724 grid points across Denmark. Figure 10 shows the CRPSS for the target month of February at

all lead times and the raw and postprocessed skill. The figure shows a reduction of the spatial variability of skill in accuracy and for this particular month, this reduction is more pronounced for $E_{T0}$. However, and as mentioned above, the reduction of spatial variability of accuracy is not enough to ensure statistical significant positive differences in skill.

We also constructed Fig. 9 but for sharpness (Fig. S12 to Fig. S14). It is evident that a loss of sharpness occurs after postprocessing in comparison to the raw forecasts for LS and QM applied to P, and QM applied to T and $E_{T0}$. Sharpness

seems to be maintained for T and $E_{T0}$ when we use the LS method. This can be explained by the fact that the correction factor applied to T forecasts is additive, which in turn changes the level of the ensemble members and has no effect in the spread of the forecasts, leaving the sharpness score equal to that of the raw forecasts. On the other hand, when the correction factor is multiplicative, as in Eq. (1) for P and $E_{T0}$, not only the level but the spread is affected. It will increase the spread when the correction factor is above 1 (which indicates an underprediction issue), and conversely, reduce the spread when the

correction factor is below 1 (indicating an overprediction issue). The larger the correction factor is the larger effect it will have in the ensemble spread. This explains why for $E_{T0}$, where biases are in general lower than biases in P, sharpness seems not to be affected. This effect is somewhat artificial and may lead to misleading evaluations of the power LS has in correcting for biases in spread.

### 3.2.3 Statistical consistency of postprocessed monthly aggregated forecasts

Second and third row in Fig. 7 and Fig. S7 to S9 show the PIT diagrams of corrected forecasts. In general, the statistical consistency seems to be improved (points closer to the 1:1 diagonal in Figure 7) to the same degree for both postprocessing methods. Although, for $E_{T0}$, this consistency is better enhanced by QM. This fact may be explained by the more evident lost of sharpness that QM has (Fig. S11 to S13) in an attempt of adjusting the biases in spread present in the raw forecasts (Fig. 7).

### 3.2.4 Accuracy of extreme precipitation and number of dry days

Figure 11 shows the skill in terms of accuracy for both monthly maximum precipitation and the monthly number of dry days. Box-plots represent the distribution of the skill score of all 662 grid cells. Skill scores are for January forecasts for all seven lead times. Two features are highlighted, first, spatial variability of skill gets reduced after postprocessing and secondly, for the skill of the number of dry days, results show that QM performs significantly better than LS. This is not surprising as QM

adjusts for biases in the whole range of percentiles of the distributions, whereas LS only focuses on the mean. Despite the reduction of the spatial variability and an increase, on average, of the skill of postprocessed monthly maximum P and number of dry days, results still show a difficulty to beat climatology, as the CRPSS is still negative, even after bias corrections are implemented.

### 4 Summary and Conclusions



The present study had two objectives. The first one was to analyze the bias and skill of the ECMWF System 4 in comparison to a climatological ensemble forecast, i.e. a forecast based on observed climatology over a period of 24 years, and well as comparing the statistical consistency between the predictive distribution and the distribution of the its verifying observations. This analysis was done for hydrological relevant variables P, T and $E_{T0}$. The conclusions of the first objective of the study and that answer the first question posed in the Sect. 1 can be summarized as follows:

- Raw seasonal forecasts of P, T and $E_{T0}$ from ECMWF System 4 exhibit biases that depend on the target month and to a lesser extent, on lead time. This result is also in accordance to what was found in Crochemore et al., (2016). There is a persistent overforecasting issue for $E_{T0}$, which combines biases of both T and incoming shortwave solar radiation.

- In addition to the biases, Crochemore et al., (2016) also found a rather similar degree of skill of the raw ECMWF System 4 forecasts for mean areal P in France. In general, skill in terms of accuracy is only present during the first month lead time, which is basically the skill of the medium-range forecast.

- One advantage ECMWF System 4 has over ensemble climatology is that forecasts are sharper. This overconfidence, combined with the biases in the mean lead to lower levels of accuracy in comparison to the accuracy of the ensemble climatological forecasts.

- Using the PIT diagrams we were able to confirm the results for the bias on the mean of P and T. Bias in spread are present for $E_{T0}$, particularly for the months with the lowest number of ensemble members (15).

The second objective was to improve the forecasts using two relatively simple methods of postprocessing: LS and QM. This was done having in mind the problems GCMs have with regards to both bias in the mean and dispersion. Modest improvements were found and can be summarized as follows:

- Both methods act equally good in removing biases in the mean.

- In terms of accuracy, mild improvements are seen on the first month lead time, especially for T and $E_{T0}$, where a higher portion of grid points are able to reach a positive skill. P and longer lead times are still difficult to improve. This may be explained by the same situation as discussed in Zhao, et al., (2017). QM assumes that there is a linear relationship between ensemble mean and observations, assumption may not hold at longer lead times reducing the effectiveness of these methods.

- Looking at the spatial distribution of skill in sharpness we see that for P, both methods tend to decrease it, with a slight increase of QM over LS. For T and $E_{T0}$, LS seems to be able to keep the sharpness of the raw forecasts. This is not the case for QM, for which for some months it manages to disappear the areas where a slight positive skill is present. Note, however, that sharpness using the LS method gets improved when the correction factor is multiplicative and less than one (positive bias; i.e., $E_{T0}$). The opposite holds, sharpness is inflated when the multiplicative correction factor is larger than one (negative bias; i.e., P). This has implications for the computation of the CRPS, as it also penalizes (rewards) for wide (narrow) ensemble forecasts, on top of the penalization for biased predictions. This situation may also explain why in Crochemore, et. al., 2016, LS has a better improvement in terms of sharpness than QM, at least for spring P.

- Statistical consistency is better improved for QM and for $E_{T0}$ forecasts that exhibit biases in the ensemble spread. QM also performs better in correcting for low values of P. This is not a surprising result, as QM corrects for biases for the entire percentile range.

We are aware that our research may have limitations. The first is that methods applied here were implemented on a grid-to-grid basis that may not correct for displacements and might lose spatio-temporal and intervariable dependencies. Spatial




correction methods have been suggested such as the ones used by Feddersen and Andersen, (2005) and Di Giuseppe et al., (2013). Another suggestion has been to recover these dependencies by adding a final postprocessing step such as the methods proposed in Clark et al., (2004) or Schefzik et al., (2013). The second is that the exclusion of postprocessing methods tailored to ensemble forecasts that take into account the joint distribution of forecasts and observations (Raftery, et.

al., 2005; Zhao et al., 2017). Their inclusion would gain a deeper insight to the comparison presented here by increasing the complexity of the correction methods and the evaluation of their added value in comparison to simpler approaches.

Postprocessing for seasonal forecasting is still a subject at its infancy, and although one could argue that advances in seasonal forecasting will make postprocessing unnecessary in the future, there is still a long way to go to get there. GCMs suffer from several issues as discussed here, however, we still encourage its use. They are physically-based, sharper than

forecasts with ensemble climatology, and once issues with their biases discussed here are fixed by means of a more realistic representation of coupled and subgrid processes and a better integration of observational data using an updated data assimilation procedure (Weisheimer and Palmer, 2014; Doblas-Reyes et al., 2013), they will be able to provide valuable information at longer lead times for sector applications such as water management.

**5 Appendix A. Probability Integral Transform Diagram**

The interpretation of the shape of the PIT diagram is based on Fig. A12 modified from (Laio and Tamea, 2007). Deviations from the 1:1 diagonal, point to the different biases in the mean and dispersion. Four situations can arise:

1. Overprediction, or positive bias in the mean: The CDF of the $z_i's$ lies above the 1:1 diagonal.

2. Underprediction, or negative bias in the mean: The CDF of the $z_i's$ lies below the 1:1 diagonal.

3. Overdispersion, or positive bias in spread (underconfident): A greater proportion of the values of the CDF lie on the

middle ranges bins of the distribution.

4. Underdispersion, or negative bias in spread (overconfidence): A greater proportion of the values of the CDF lie on the tails of the distribution.

**6 Acknowledgements**

This study was supported by the project 'HydroCast - Hydrological Forecasting and Data Assimilation', Contract #0603-

00466B (http://hydrocast.dhigroup.com/) funded by the Innovation Fund Denmark. Special thanks to Florian Pappenberger for providing the ECMWF System 4 reforecast and to Andy Wood and Pablo Mendoza for hosting the first author at NCAR.

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

## 8 Figure Captions

**Figure 1:** Spatio-temporal variability of precipitation, temperature and reference evapotranspiration of monthly aggregated values (P, $E_{T0}$) and monthly averages (T) of the observation period (1990-2013).

**Figure 2:** Example of a monthly forecast, valid for August and lead time 1 month for a grid point located in west-central Denmark. Blue shaded box-plots are ensemble climatological forecasts. Black box-plots are ECMWF System 4 raw or postprocessed forecasts. Red dots represent observed values. Bias as in Eq. (4) and Eq. (5) and skill scores for accuracy and sharpness (CRPSS and SS) as in Eq. (8). First column corresponds to the raw forecast, second and third columns correspond to the corrected forecasts with the Linear Scaling/Delta Change (LS) and Quantile Mapping (QM) methods, respectively.

**Figure 3:** Percentage bias and absolute of monthly values of raw forecasts. Y-axis represents the target month and the X-axis represents the different lead times at which target months are forecasted. Values in blue range represent a positive bias and values in red represent a negative bias.

**Figure 4:** Percentage bias and absolute bias of monthly values of raw forecasts for the summer. Forecast lead time of 1 month.

**Figure 5:** Skill in terms of sharpness and accuracy of monthly values of raw forecasts. Y-axis represents the target month and X-axis represents the different lead times at which target month is forecasted. Changes in color represent different skill in accuracy levels and different shapes represent difference in sharpness. Grey color means that the skill in terms of accuracy and squared shape means that the skill in terms of sharpness are respectively worse than the ensemble climatological forecasts.

**Figure 6:** Spatial variability of skill in accuracy for summer raw forecasts for lead time 1 month. The grids marked with '*' are points where the distribution of the accuracy for ensemble climatology differs from the accuracy distribution of the ECMWF System 4 forecasts at a 5% significance using the WMW-test.

**Figure 7:** PIT diagrams of summer P, T and $E_{T0}$ for the raw and postprocessed forecasts for lead time 1 month. The interpretation of the PIT diagram is explained in Sect. 5 and Fig. A12.

**Figure 8:** Biases of raw and postprocessed P, T and $E_{T0}$ at four locations in Denmark. Biases are for the different target months and for lead time 1 month.

**Figure 9:** Percentage of grid points with statistically significant positive CRPSS cf. Eq. (8).





**Figure 10:** Spatial variability of skill in terms of accuracy for P, T and $E_{T0}$ for the raw and post-processed forecasts of February as a target month at different lead times. Box-plots represent the values of CRPSS cf. Eq. (8) with climatology as reference, of all the 662/724 grid points covering Denmark.

**Figure 11:** Skill of daily monthly maximum P and number of dry days for target month January for all 7-month lead times for the raw and post-processed forecasts. Box-plots represent the values of CRPSS, of Eq. (8) with climatology as reference, of all the 662/724 grid points covering Denmark.

**Figure A12:** Probability Integral Transform (PIT) Diagram. Modified from (Laio and Tamea, 2007). Observed values are generated from a standard normal distribution $N(0,1)$. Forecasts for these observations are then generated as follows: $N(1.5,1)$ for overestimation, $N(-1.5,1)$ for underestimation, $N(0,3)$ for an overdispersive system and $N(0.0.3)$ for a underdispersive system.





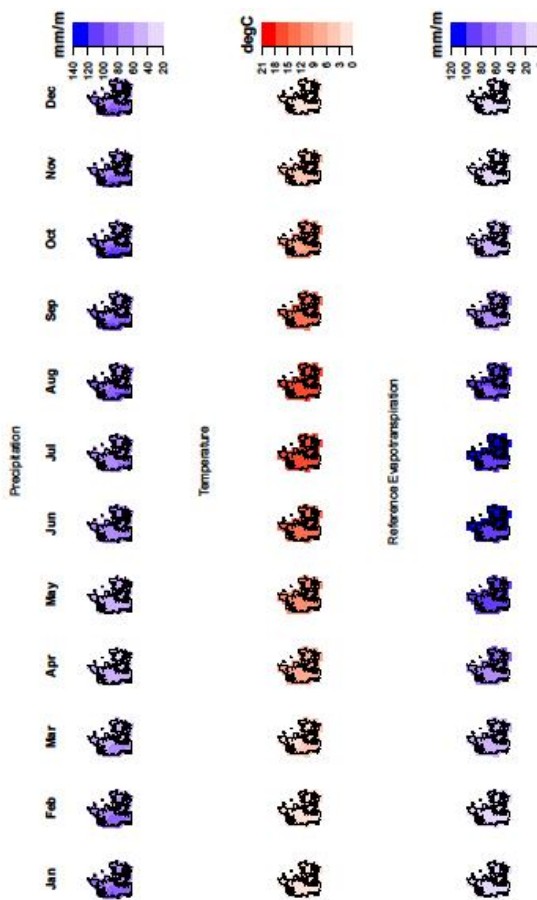

**Figure 1:** Spatio-temporal variability of precipitation, temperature and reference evapotranspiration of monthly aggregated values (P, $E_{T0}$) and monthly averages (T) of the observation period (1990-2013).

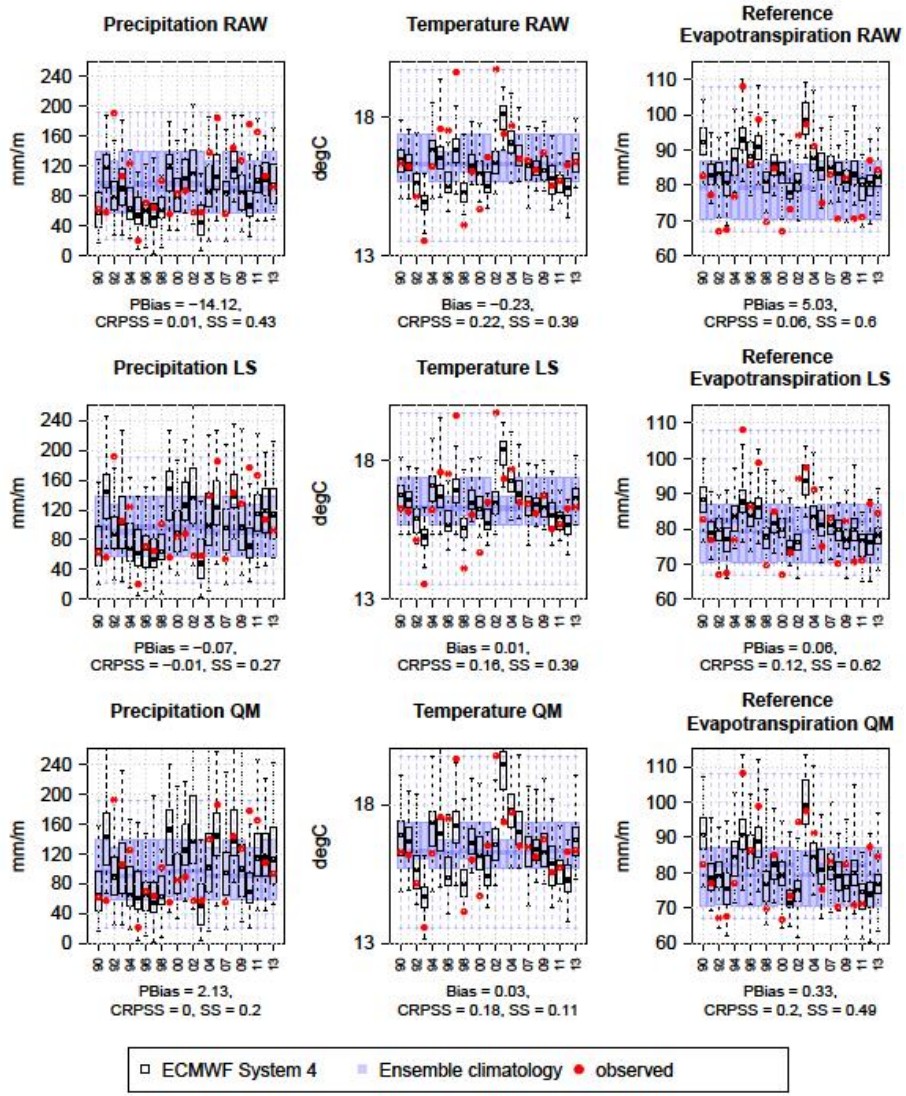

**Figure 2:** Example of a monthly forecast, valid for August and lead time 1 month for a grid point located in west-central Denmark. Blue shaded box-plots are ensemble climatological forecasts. Black box-plots are ECMWF System 4 raw or postprocessed forecasts. Red dots represent observed values. Bias as in Eq. (4) and Eq. (5) and skill scores for accuracy and sharpness (CRPSS and SS) as in Eq. (8). First column corresponds to the raw forecast, second and third columns correspond to the corrected forecasts with the Linear Scaling/Delta Change (LS) and Quantile Mapping (QM) methods, respectively.





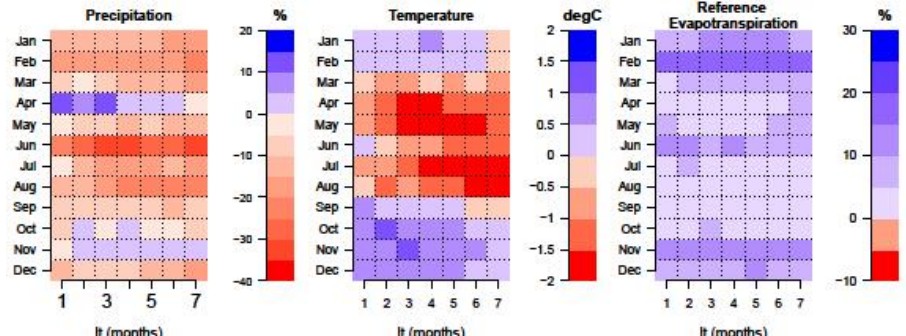

**Figure 3:** Percentage bias and absolute of monthly values of raw forecasts. Y-axis represents the target month and the X-axis represents the different lead times at which target months are forecasted. Values in blue range represent a positive bias and values in red represent a negative bias.





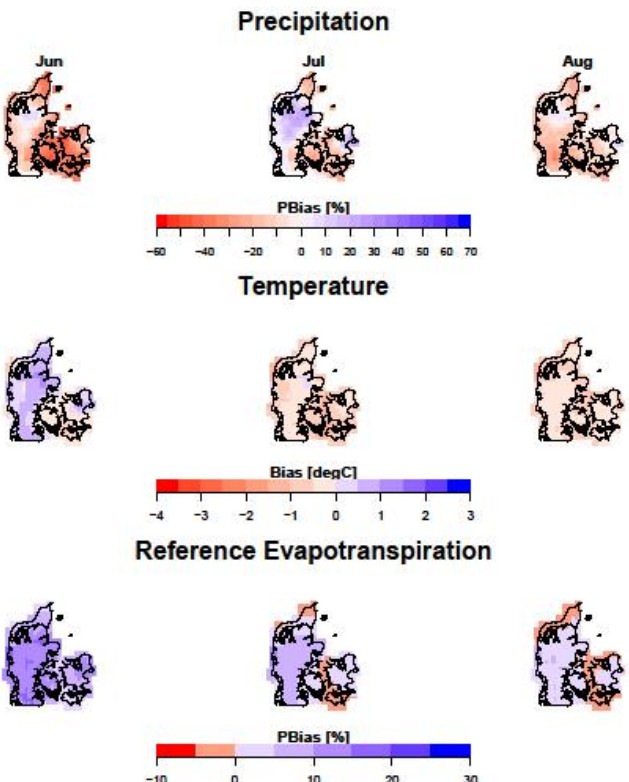

**Figure 4:** Percentage bias and absolute bias of monthly values of raw forecasts for the summer. Forecast lead time of 1 month.





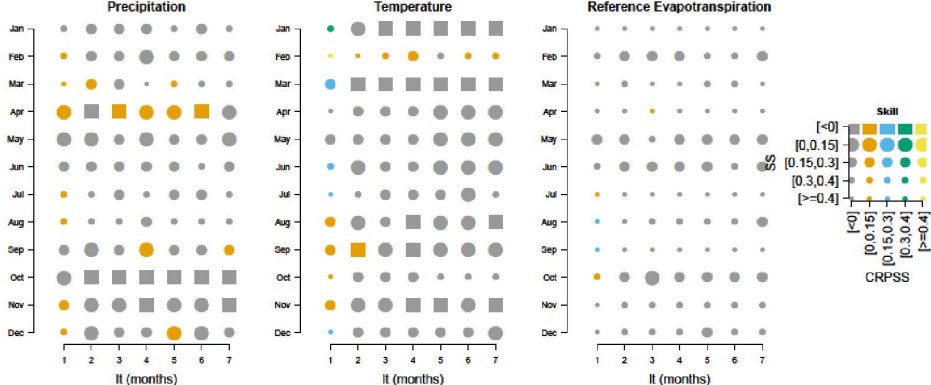

**Figure 5:** Skill in terms of sharpness and accuracy of monthly values of raw forecasts. Y-axis represents the target month and X-axis represents the different lead times at which target month is forecasted. Changes in color represent different skill in accuracy levels and different shapes represent difference in sharpness. Grey color means that the skill in terms of accuracy and squared shape means that the skill in terms of sharpness are respectively worse than the ensemble climatological forecasts.



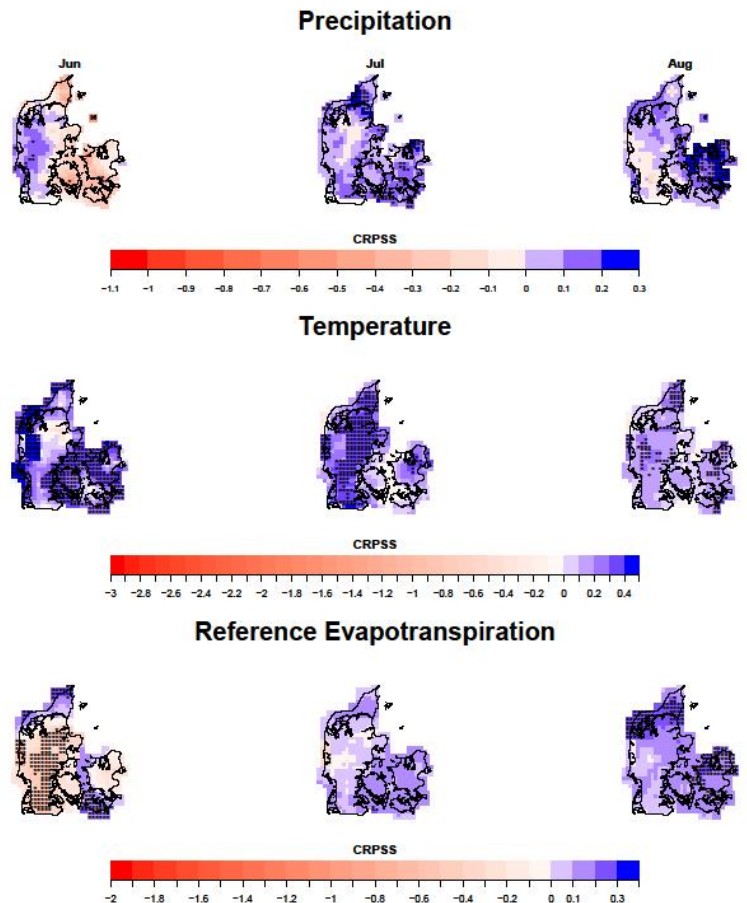

**Figure 6:** Spatial variability of skill in accuracy for summer raw forecasts for lead time 1 month. The grids marked with '*' are points where the distribution of the accuracy for ensemble climatology differs from the accuracy distribution of the ECMWF System 4 forecasts at a 5% significance using the WMW-test.





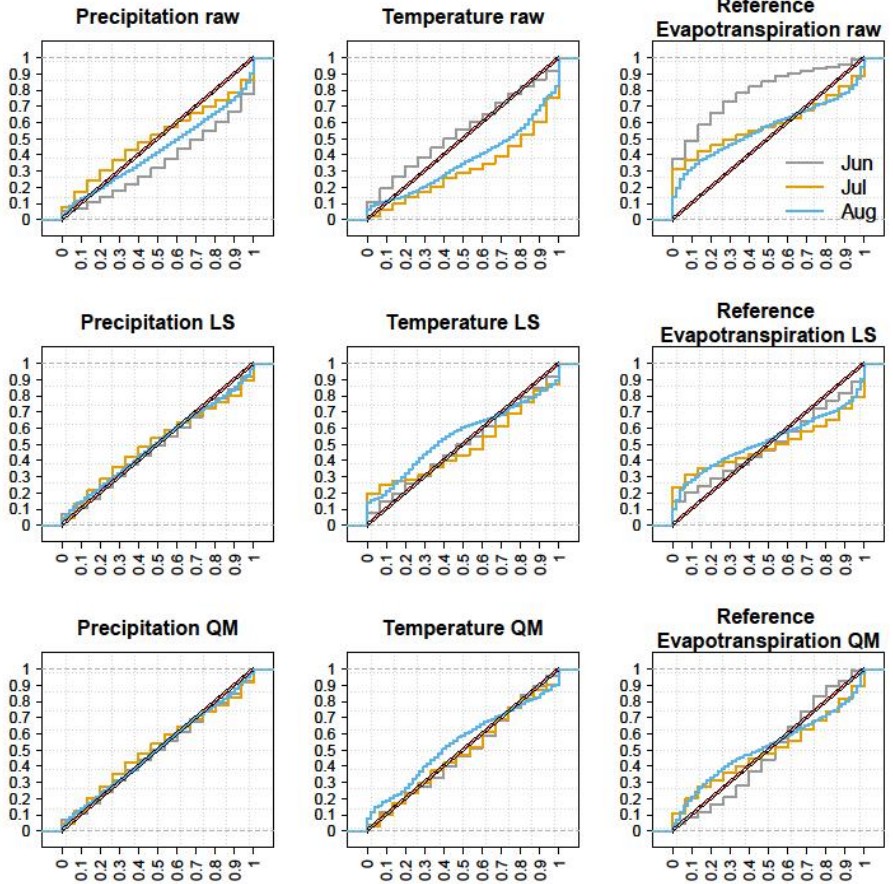

**Figure 7:** PIT diagrams of summer P, T and $E_{T0}$ for the raw and postprocessed forecasts for lead time 1 month. The interpretation of the PIT diagram is explained in Sect. 5 and Fig. A12.



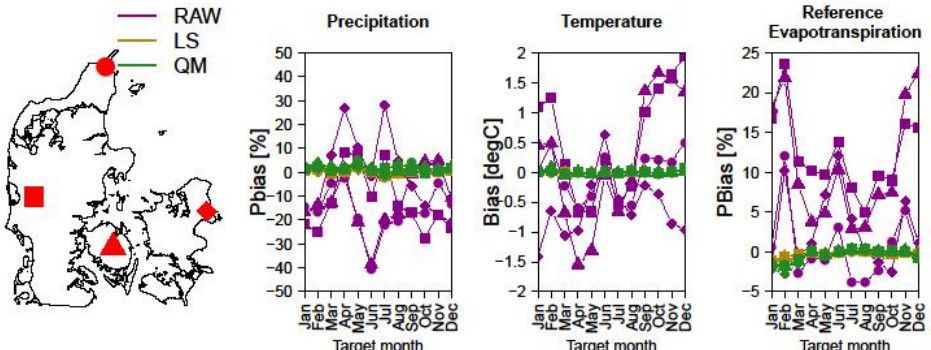

**Figure 8:** Biases of raw and postprocessed P, T and $E_{T0}$ at four locations in Denmark. Biases are for the different target months and for lead time 1 month.

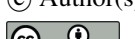



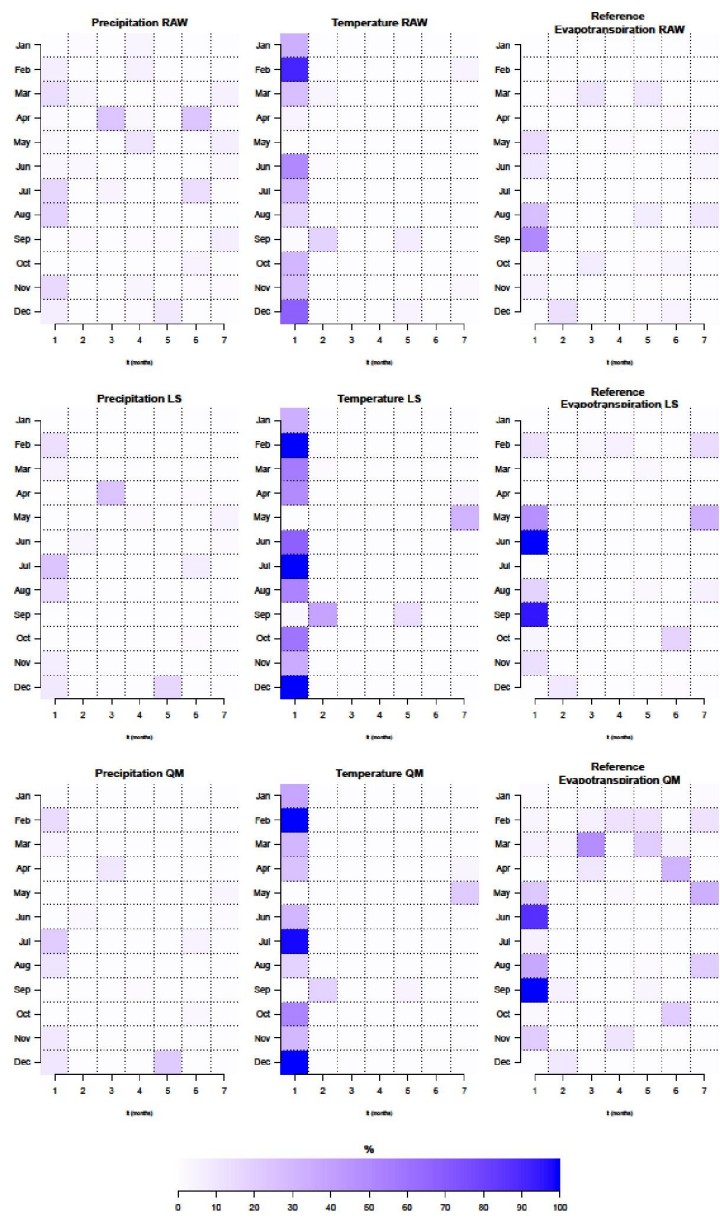

**Figure 9:** Percentage of grid points with statistically significant positive CRPSS cf. Eq. (8).





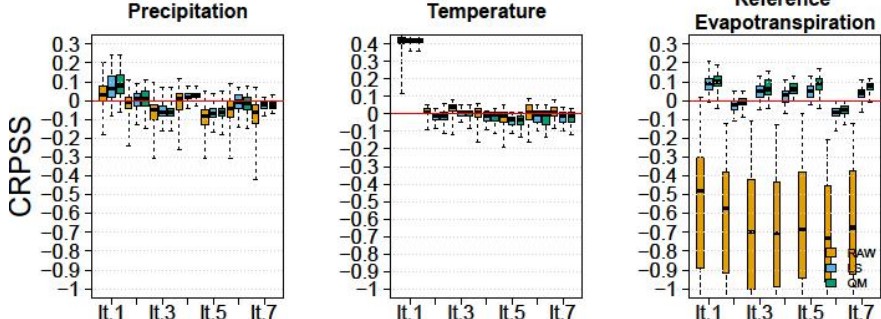

**Figure 10:** Spatial variability of skill in terms of accuracy for P, T and $E_{T0}$ for the raw and post-processed forecasts of February as a target month at different lead times. Box-plots represent the values of CRPSS cf. Eq. (8) with climatology as reference, of all the 662/724 grid points covering Denmark.





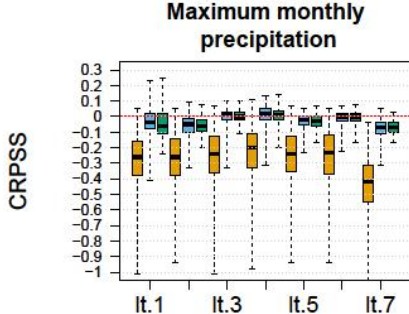
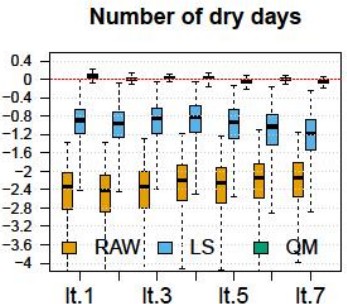

**Figure 11:** Skill of daily monthly maximum P and number of dry days for target month January for all 7-month lead times for the raw and post-processed forecasts. Box-plots represent the values of CRPSS, of Eq. (8) with climatology as reference, of all the 662/724 grid points covering Denmark.





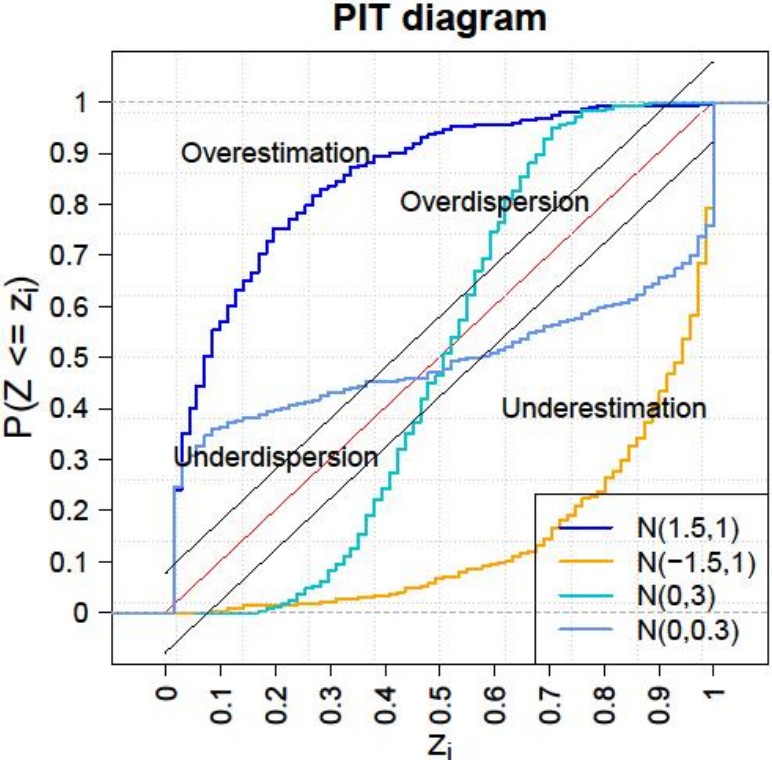

**Figure A12:** Probability Integral Transform (PIT) Diagram. Modified from (Laio and Tamea, 2007). Observed values are generated from a standard normal distribution $N(0,1)$. Forecasts for these observations are then generated as follows: $N(1.5,1)$ for overestimation, $N(-1.5,1)$ for underestimation, $N(0,3)$ for an overdispersive system and $N(0.0.3)$ for a underdispersive system.