# Peer review of "On the skill of raw and postprocessed ensemble seasonal meteorological forecasts in Denmark"

_Hydrology and Earth System Sciences, 2017_

## Referee Comment (RC1) · Anonymous Referee #1 · 7 Aug 2017

This paper investigates the performance of precipitation and temperature forecasts from ECMWF System 4, as well as derived reference evapotranspiration. The authors also look at the impact of two simple postprocessing methods: linear scaling and quantile mapping, on the performance of these forecasts. Raw forecasts tended to be overconfident, and regions with biases also corresponded to regions with lower skill. Both linear scaling and quantile mapping performed well in removing biases, quantile mapping was better suited to improve statistical consistency.

General comment

I found the paper well-written and I think that it provides both didactic explanations for the methodology as well as an in-depth and comprehensive analysis of the skill and bias patterns. It also fits in the Subseasonal-to-seasonal special issue since it follows

and complements nicely the paper by Crochemore et al. (2016) published in this same issue. I list below a few comments about this version of the paper. These comments are mainly technical. My main remark would be that the figures are generally too small and thus difficult for the reader to read and interpret. I detail this point further down.

Major comments and general questions

Section 2.2: What do you think is the impact of the interpolation method on the results and areas with skill? How common is the inverse distance weighting to interpolate meteorological variables? Couldn't this induce yet another bias in the forecasts? If observations had been upscaled to preserve the scale of the GCM forecasts, would you expect similar results?

Section 3.2.4 and Figure 11: It was unclear to me why linear scaling impacted the number of dry days. If dry days are defined as days with no precipitation, and linear scaling solely consists in applying a multiplicative factor, the number of zero-values should not be affected. Did you define a threshold to determine dry days? Please clarify this.

Technical comments

P2 L7: Replace "Despite of the efforts" by "Despite the efforts".

P2 L35: I suggest adding "only" after "for precipitation", for clarification.

P4 L2-4: Please check the indices in equations 1 and 2. It seems that index i is used to represent different things: the year for which the correction is applied and a sum that runs from 1 to N while excluding the year for correction itself (previously represented by i).

P4 L6: If I understand correctly, N is equal to the numbers you have, i.e. 24 years from 1990 to 2013. Is that correct?

P8 L35: Replace "thorough" with "through".
P9 L6-8: Could you please clarify this point?

P9 L11: Replace "The second and third columns column" by "The second and third rows"

P10 L27-29: "This fact may be . . . in the raw forecasts (Fig. 7)" could you please reformulate this sentence?

P11 L13: It seems that in this context and given the following sentence, "sharpness" can hardly be an advantage.

P11 L21: I suggest replacing "act equally good" by "perform equally well".

P11 L27-30: Please reformulate these sentences.

P12 L3: Replace "The second is that the exclusion" with "The second is the exclusion".

Figures 1, 4 and 6 and maps in the Supplement: The maps are too small to easily distinguish the patterns. In addition, it is difficult to spot the stars in Figure 6, both due to the size and the colors. I suggest making the maps bigger, and if necessary, changing the color of the stars in Figure 6.

Figure 2: Please explain the x-axis somewhere or make the years fully explicit.

Figure 3: The x-axis is not the same size in all three graphs. The size used in the left-hand graph is easier to read.

Figure 5: Please increase the size of the axis labels. Consider replacing "lt" by "lead times". Please also reformulate the last sentence in the legend.

Figure 9 and similar graphs in the Supplement: Please also increase the labels here.

Figure 10: I recommend moving the legend to the first or second graph for readability.

Figure A12: I think N(0.0.3) should be N(0,0.3)
* * *
[Figure]

366, 2017.

---

## Referee Comment (RC2) · Anonymous Referee #2 · 11 Sep 2017

The paper is well written and the results could be of interest for researchers testing extended range and seasonal forecasts especially for hydrological applications. Therefore the paper is worth to be published after some minor corrections and/or inclusions of additional explanations. 1. On page 3 you describe the ECMWF forecast data. I was wondering why there some months with 15 and some with 51 members? Could you explain this? Did you include some corrections in the skill scores for the 15 ensembles (e.g. Müller, W.A., C. Appenzeller, F.J. Doblas-Reyes, and M.A. Liniger, 2005: A Debiased Ranked Probability Skill Score to Evaluate Probabilistic Ensemble Forecasts with Small Ensemble Sizes. J. Climate, 18, 1513–1523, https://doi.org/10.1175/JCLI3361.1) in order to make them comparable with the 51 members? Have there been any model changes within these 24 years? If yes, has

this been taken into consideration? 2. Also on page 3 line 34 you give the dimension of 662x12x7x24. Does this mean grid cells x months x forecasts x years ? Could you please clarify this? 3. On page 4 you explain the the QM approach. In line 16 you write that the EDF has been trained. I would rather call this process fitting. 4. Regarding the skill (from page 5 onwards): The CRPS is a global score combining the reliability and sharpness aspects. You mention that you use the CRPS as a general measure of accuracy, but you don't say why it is general. I think this should be included for readers who are not familiar with the CRPS (e.g. Gneiting, T., A. Raftery, A. Westveld III, and T. Goldman (2005), Calibrated probabilistic forecasting using ensemble model output statistics and minimum CRPS estimation, Mon. Weather Rev., 133(5), 1098–1118) 5. Have you tried to eliminate the drizzle effects and include different thresholds for zero precipitation? This could be interesting for analysing the dry periods. 6. I agree with Reviewer 1 that the Figures are difficult to read. Regarding Figure 2 it would be interesting to compare the boxplots of the forecasts with boxplots of the climatology in order to see the median, interquartile range. 7. I don't think that the PIT diagram has to be explained in that detail and you could delete Appendix A and Figure A12. You can find the same Figures in Laio, F., and S. Tamea (2007), Verification tools for probabilistic forecasts of continuous hydrological variables, Hydrol. Earth Syst. Sci., 11(4), 1267–1277 and in Thyer, M., B. Renard, D. Kavetski, G. Kuczera, S. W. Franks, and S. Srikanthan (2009), Critical evaluation of parameter consistency and predictive uncertainty in hydrological modeling: A case study using Bayesian total error analysis, Water Resour. Res., 45, W00B14, doi:10.1029/2008WR006825.

---

## Referee Comment (RC3) · Anonymous Referee #3 · 18 Sep 2017

**1   General comments**

The paper by Lucatero et al. describes an assessment of uncorrected and post-processed GCM forecast over Denmark during the period 1990-2013. The study adresses a critical issue for GCM forecast users, especially in the field of hydrology where uncorrected forecast often lacks sufficient skill to be used as input for hydrological applications. Most methods applied in the paper are sound, for example the use of two well established post-processing techniques and a leave-out cross validation scheme. The paper is well written with a clear and concise structure. However, we believe that there is scope to improve its content before it can be published. Our major comments are listed below:

[Figure]

1. The approach that was adopted by the authors to downscale the ECMWF forecast is questionable. The authors applied an inverse weighting algorithm to convert the 70km resolution ECMWF grid to a 10 km resolution, and then used the downscaled data to post-process and analyse forecast performance. By doing this, they smooth ECMWF rainfall surfaces and break the conservation of mass, which artificially reduces the skill of uncorrected forecasts. To circumvent this problem, we recommend performing the analysis undertaken by the authors at the resolution of ECMWF forecasts (i.e. 70km), using a simple aggregation method for gridded observation data (see for example Schepen et al. (2014)). This alternative approach would eliminate the need for a downscaling algorithm, and provide a direct assessment of uncorrected ECMWF forecasts compared to post-processed forecasts.

   We understand the value of downscaling to work at a meaningful scale for hydrological applications. However, downscaling is a research topic in its own, and its impact should not mask the skill of the uncorrected forecasts. Without a proper assessment of uncorrected forecasts, it is difficult to select the appropriate downscaling model.

2. The quality and resolution of several figures is clearly below the standard of an international scientific journal. We strongly recommend redrawing figures 1, 4 and 6, increasing the resolution and/or converting them to a vector format. Unfortunately, with such low figure quality, it becomes difficult to check the comments made by the authors in reference to those figures. Additional comments on the figures are provided in the next section to improve their readability.

3. The analysis of forecast reliability (or statistical consistency as per the author's nomenclature) lacks important information to properly assess forecast performance:

   - It is not clear which variables are used to draw the PIT plots (figure 7). Such

plots require a single series of PIT values computed from matched pairs of observations and forecasts. The authors do not precise if the observations and forecasts are coming from a single grid cell, or from a spatial aggregation (e.g. the whole Denmark). This point is important to understand their difficulties in interpreting the PIT plot (see Line 6 page 9:"issues (...) are not remarkably clear"). We suggest drawing the PIT plots for a selected set of grid points and one lead time representing the main characteristics of the PITs across the study area.

- The analysis of reliability lacks a quantitative criterion similar to the skill scores. A standard approach is to compute the pvalue of a uniformity test such as the Kolmogorov-Smirnov test (Laio and Tamea, 2007). However, instead of the KS test, we recommend the Anderson-Darling test (Anderson and Darling, 1952), which exhibits a greater power (Noceti et al., 2003) and better ability to detect deviations from uniformity of the extremes. We strongly suggest computing the p-value of such tests for all grid points of the domain and all lead times, and then summarise the results by counting how many cells pass the test with a 5% threshold for a given lead time. This will provide a consistent and reproducible assessment of forecast reliability across the study area.

4. Temperature is treated differently than rainfall and PET both in the computation of the bias score and in the implementation of the LS post-processing model. This is quite confusing and should be harmonised. We recommend that all bias scores be computed in relative terms to facilitate comparison. The LS model should be applied to the three variables in two modes: multiplicative (which is the configuration used for P and PET) and additive (used for T). This approach would provide a clear advice on the choice of the LS configuration.

5. It is not clear how extrapolation is undertaken within the QM model. The authors only indicate that "Extrapolation is then needed to map ensemble values and

percentiles that are outside the training range" (see Page 4, Line 21). This is a critical and frequent problem with the QM method which affects extremes, and is then of particular interest in hydrological forecasts. Please detail the extrapolation method.

Considering that the paper is covering an important topic for seasonal forecasting, but that the number of points to be improved is quite significant, we recommend the paper to be accepted with major revision. Detailed comments are provided in the following section.

**2   Specific comments**

1. Page 2, Line 26, "The most used methods are linear scaling and quantile mapping": Zhao et al. (2017) provides an interesting perspective on the limitations of Quantile-Quantile mapping that could be cited here.

2. Page 3, Line 4, "A statistical consistent forecast system has low (or non-existent) bias in both mean and variance": correct? ok.

3. Page 3, Line 14, "we make use of the Makkink equation (Hendriks, 2010) that takes as inputs temperature and incoming short-wave solar radiation from ECMWF System 4": a mention on the quality of radiation forecast would be useful. A full analysis of radiation forecast performance is out of scope here, but perhaps some references can be cited to understand the impact of radiation inputs in ET0 calculation.

4. Page 3, Line 20, "The time and spatial variations of the variables can be seen in Fig. 1.": Fig 1 is not readable, so the accuracy of this comment is hard to assess.

5. Page 3, Line 38, "once a scale factor has been applied": In the case of T, the factor is not a scale, but a shift.

6. Page 6, Line 2, "a Wilcoxon-Mann-Whitney test was carried out.": What was the data used to apply the WMW test? Was it applied to all grid cells for a single lead time? Please clarify.

7. Page 6, Line 39, "April shows an overestimation that might be due to the 'drizzle effect' in a month where dry days are more common.": It is quite surprising that this "drizzle effect" is not affecting the bias in March and May where the bias shows an opposite trend compared to April. This statement requires more explanations. We suggest showing the number of dry days per month, to confirm that April has the highest proportion.

8. Page 11, Line 13, "One advantage ECMWF System 4 has over ensemble climatology is that forecasts are sharper": The forecast is sharper but it is not clear if it is reliable. In this case, being sharp is not an advantage, but a problem.

9. Page 11, Line 24, "QM assumes that there is a linear relationship between ensemble mean and observations, assumption may not hold": We believe that the authors mean LS instead of QM here. Otherwise we do not see why QM would assume such linear relationship.

10. Figure 5: The figure is trying to convey too many information at the same time with a confusing choice of symbol sizes (decreasing symbol size for better sharpness skill score) and color schemes. We suggest using the same presentation than Figure 3 and grouping the two figures into a single one that would provide an homogenous overview of forecast performance.

11. Figure 6: The scale of the color bar is different between the 3 plots. As a result, it is impossible to compare the skill between the three variables.

12. Figure 8: This plot is difficult to read because the legend does not explain that each location has a different symbol and all curves are shown with the same color. We suggest splitting each one of the three plots into 4 different subplots showing the results for one location only.

**References**

Anderson, T. W. and Darling, D. A. (1952). Asymptotic theory of certain" goodness of fit" criteria based on stochastic processes. *The annals of mathematical statistics*, pages 193–212.

Laio, F. and Tamea, S. (2007). Verification tools for probabilistic forecasts of continuous hydrological variables. *Hydrology and Earth System Sciences*, 11(4):1267–1277.

Noceti, P., Smith, J., and Hodges, S. (2003). An evaluation of tests of distributional forecasts. *Journal of Forecasting*, 22(6-7):447–455.

Schepen, A., Wang, Q., and Robertson, D. E. (2014). Seasonal forecasts of australian rainfall through calibration and bridging of coupled gcm outputs. *Monthly Weather Review*, 142(5):1758–1770.

Zhao, T., Bennett, J. C., Wang, Q., Schepen, A., Wood, A. W., Robertson, D. E., and Ramos, M.-H. (2017). How suitable is quantile mapping for postprocessing gcm precipitation forecasts? *Journal of Climate*, 30(9):3185–3196.

---

## Author Comment (AC1) · 12 Dec 2017

We appreciate the time spent carefully reviewing this manuscript. We are certain that your comments and suggestions will improve the quality of this paper.

**Anonymous Referee #1**

**This paper investigates the performance of precipitation and temperature forecasts from ECMWF System 4, as well as derived reference evapotranspiration. The authors also look at the impact of two simple postprocessing methods: linear scaling and quantile mapping, on the performance of these forecasts. Raw forecasts tended to be overconfident, and regions with biases also corresponded to regions with lower skill. Both linear scaling and quantile mapping performed well in removing biases, quantile mapping was better suited to improve statistical consistency. General comment I found the paper well-written and I think that it provides both didactic explanations for the methodology as well as an in-depth and comprehensive analysis of the skill and bias patterns. It also fits in the Subseasonal-to-seasonal special issue since it follows and complements nicely the paper by Crochemore et al. (2016) published in this same issue.**

**I list below a few comments about this version of the paper. These comments are mainly technical. My main remark would be that the figures are generally too small and thus difficult for the reader to read and interpret. I detail this point further down.**

**Major comments and general questions**

**Section 2.2: What do you think is the impact of the interpolation method on the results and areas with skill? How common is the inverse distance weighting to interpolate meteorological variables? Couldn't this induce yet another bias in the forecasts? If observations had been upscaled to preserve the scale of the GCM forecasts, would you expect similar results?**

We made the decision of the interpolation procedure based on its simplicity and on studies that have used a similar method (Wood et. al., 2002; Voision et al., 2010; Tian, 2014). We are aware of the issues that come up when using such an interpolation method. However, the disadvantages of this choice in comparison to others (spatial interpolation, upscale observations, statistical downscaling) is a study of its own and out of the scope of this manuscript. Besides, computing aerial precipitation it is not enough for our purposes. We needed to investigate the behavior of the forecasts on a resolution relevant to hydrological forecasting. Additionally, we do not believe that, at least for monthly accumulated values, values of skill in accuracy will change in a significant manner. Evidence for the latter claim can be found in Fig. 1 below where, as you suggest, the comparison is done on the 70 km grid box versus inside box average.

[Figure]

**Fig 1.** Skill in terms of accuracy of monthly values of raw forecasts at the 10 km grid (first line) and 70 km grid (second line). Y-axis represents the target month and X-axis represents the different lead times at which target month is forecasted.

**Section 3.2.4 and Figure 11: It was unclear to me why linear scaling impacted the number of dry days. If dry days are defined as days with no precipitation, and linear scaling solely consists in applying a multiplicative factor, the number of zero-values should not be affected. Did you define a threshold to determine dry days? Please clarify this.**

We failed to note in Sect. 2.3.1 that a previous step was introduced before applying the correction factor for precipitation. An analysis was made as to the threshold for which the number of dry days in a given month was equal to the observed number of dry days. On average along Denmark, this value is of 1.5 mm/day. That is why in Sect. 3.2.4 and in Fig. 11, LS seems to increase the skill in predicting the number of dry days. It is then a consequence of this threshold rather than the method itself. Note that this previous step was not introduced in QM as it will map small forecast precipitation values with larger observed values. We will make this clearer in the new version of the manuscript.

**Technical comments**

**P2 L7: Replace "Despite of the efforts" by "Despite the efforts".**

Yes, will do.

**P2 L35: I suggest adding "only" after "for precipitation", for clarification.**

We will change this accordingly.

**P4 L2-4: Please check the indices in equations 1 and 2. It seems that index i is used to represent different things: the year for which the correction is applied and a sum that runs from 1 to N while excluding the year for correction itself (previously represented by i).**

You are correct. Thanks for noticing this, we will correct it.

**P4 L6: If I understand correctly, N is equal to the numbers you have, i.e. 24 years from 1990 to 2013. Is that correct?**

As the previous notation, it will be N-1.

**P8 L35: Replace "thorough" with "through".**

**P9 L6-8: Could you please clarify this point?**

There are two features at play that might prevent us from detecting biases in ensemble spread with the presented configuration of the PIT diagrams:

(1) The spatial pooling of the $z_i$.

(2) The monthly accumulation of the variables as stated in P9 L6-8.

In the updated version of the manuscript we will address the effect of each feature to clarify this point as also suggested by Reviewer # 3 in comment 3a.

**P9 L11: Replace "The second and third columns column" by "The second and third rows"**

Yes, will do.

**P10 L27-29: "This fact may be ... in the raw forecasts (Fig. 7)" could you please reformulate this sentence?**

Yes, will do.

**P11 L13: It seems that in this context and given the following sentence, "sharpness" can hardly be an advantage.**

We will reformulate the sentence accordingly.

**P11 L21: I suggest replacing "act equally good" by "perform equally well".**

Yes, will do.

**P11 L27-30: Please reformulate these sentences.**

Yes, will do.

**P12 L3: Replace "The second is that the exclusion" with "The second is the exclusion".**

We are aware of the issue with the plots, we will improve the layout of all figures as you carefully suggested.

**Figures 1, 4 and 6 and maps in the Supplement: The maps are too small to easily distinguish the patterns. In addition, it is difficult to spot the stars in Figure 6, both due to the size and the colors. I suggest making the maps bigger, and if necessary, changing the color of the stars in Figure 6.**

**Figure 2: Please explain the x-axis somewhere or make the years fully explicit.**

**Figure 3: The x-axis is not the same size in all three graphs. The size used in the left-hand graph is easier to read.**

**Figure 5: Please increase the size of the axis labels. Consider replacing "lt" by "lead times". Please also reformulate the last sentence in the legend.**

**Figure 9 and similar graphs in the Supplement: Please also increase the labels here.**

**Figure 10: I recommend moving the legend to the first or second graph for readability.**

**Figure A12: I think N(0.0.3) should be N(0,0.3).**

**Reviewer # 2**

**The paper is well written and the results could be of interest for researchers testing extended range and seasonal forecasts especially for hydrological applications. Therefore the paper is worth to be published after some minor corrections and/or inclusions of additional explanations.**

**1. On page 3 you describe the ECMWF forecast data. I was wondering why there some months with 15 and some with 51 members? Could you explain this?**

**Did you include some corrections in the skill scores for the 15 ensembles (e.g. Müller, W.A., C. Appenzeller, F.J. Doblas-Reyes, and M.A. Liniger, 2005: A Debiased Ranked Probability Skill Score to Evaluate Probabilistic Ensemble Forecasts with Small Ensemble Sizes. J. Climate, 18, 1513–1523, https://doi.org/10.1175/JCLI3361.1) in order to make them comparable with the 51 members?**

**Have there been any model changes within these 24 years? If yes, has this been taken into consideration?**

First, this is the data we received from the meteorological forecast provider (ECMWF). We assume that the increase of ensemble size for February, May, August and November is done with the objective of increasing quality of the forecasts of the upcoming season, for example summer (JJA) for forecasts initialized in May. We will clarify if this is the case in the updated version of the manuscript.

Second, we did not include corrections to the estimated CRPS as in Ferro et al., (2008) and references therein. We will evaluate whether this correction is applicable to our case and apply the correction to obtain unbiased estimator of CRPS as done in Crochemore et al., (2016). If we, however, consider that the assumptions of such estimators are not met (i.e., perfect reliability) we will explain our decision and discuss it the new version of the manuscript.

Finally, to the knowledge of the authors, there was no model update in the 24 year period used in the present manuscript.

**2. Also on page 3 line 34 you give the dimension of 662x12x7x24. Does this mean grid cells x months x forecasts x years ? Could you please clarify this?**

It means grid cells, months, lead times, years. This will be clarified in the revised manuscript.

**3. On page 4 you explain the the QM approach. In line 16 you write that the EDF has been trained. I would rather call this process fitting.**

Yes, you are correct. We will correct the manuscript accordingly.

**4. Regarding the skill (from page 5 onwards): The CRPS is a global score combining the reliability and sharpness aspects. You mention that you use the CRPS as a general measure of accuracy, but you don't say why it is general. I think this should be included for readers who are not familiar with the CRPS (e.g. Gneiting, T., A. Raftery, A. Westveld III, and T. Goldman (2005), Calibrated probabilistic forecasting using ensemble model output statistics and minimum CRPS estimation, Mon. Weather Rev., 133(5), 1098–1118)**

We will improve our explanation of the CRPS for better clarity in the next version of the manuscript.

**5. Have you tried to eliminate the drizzle effects and include different thresholds for zero precipitation? This could be interesting for analysing the dry periods.**

We failed to note in Sect. 2.3.1 that a previous step was introduced before applying the correction factor for precipitation. An analysis was made as to the threshold for which the number of dry days in a given month was equal to the observed number of dry days. On average along Denmark, this value is of 1.5 mm/day. That is why in Sect. 3.2.4 and in Fig. 11, LS seems to increase the skill in predicting the number of dry days. It is then a consequence of this threshold rather than the method itself. Note that this previous step was not introduced in QM as it will map small forecast precipitation values with larger observed values. We will make this clearer in the new version of the manuscript.

**6. I agree with Reviewer 1 that the Figures are difficult to read. Regarding Figure 2 it would be interesting to compare the boxplots of the forecasts with boxplots of the climatology in order to see the median, interquartile range.**

We will improve the layout of all figures in the updated version of the manuscript. In Fig. 2 you can see the boxplots of the forecasts (black) and the boxplots of climatology (light blue) as you suggest.

**7. I don't think that the PIT diagram has to be explained in that detail and you could delete Appendix A and Figure A12. You can find the same Figures in Laio, F., and S. Tamea (2007), Verification tools for probabilistic forecasts of continuous hydrological variables, Hydrol. Earth Syst. Sci., 11(4), 1267–1277 and in Thyer, M., B. Renard, D. Kavetski, G. Kuczera, S. W. Franks, and S. Srikanthan (2009), Critical evaluation of parameter consistency and predictive uncertainty in hydrological modeling: A case study using Bayesian total error analysis, Water Resour. Res., 45, W00B14, doi:10.1029/2008WR006825.**

We will remove the Appendix and Figure A12 and refer to the papers you mention instead.

**Reviewer # 3**

**1 General comments**

**The paper by Lucatero et al. describes an assessment of uncorrected and postprocessed GCM forecast over Denmark during the period 1990-2013. The study adresses a critical issue for GCM forecast users, especially in the field of hydrology where uncorrected forecast often lacks sufficient skill to be used as input for hydrological applications. Most methods applied in the paper are sound, for example the use of two well established post-processing techniques and a leave-out cross validation scheme. The paper is well written with a clear and concise structure. However, we believe that there is scope to improve its content before it can be published. Our major comments are listed below:**

**1. The approach that was adopted by the authors to downscale the ECMWF forecast is questionable. The authors applied an inverse weighting algorithm to convert the 70km resolution ECMWF grid to a 10 km resolution, and then used the downscaled data to post-process and analyse forecast performance. By doing this, they smooth ECMWF rainfall surfaces and break the conservation of mass, which artificially reduces the skill of uncorrected forecasts. To circumvent this problem, we recommend performing the analysis undertaken by the authors at the resolution of ECMWF forecasts (i.e. 70km), using a simple aggregation method for gridded observation data (see for example Schepen et al. (2014)). This alternative approach would eliminate the need for a downscaling algorithm, and provide a direct assessment of uncorrected ECMWF forecasts compared to post-processed forecasts. We understand the value of downscaling to work at a meaningful scale for hydrological applications. However, downscaling is a research topic in its own, and its impact should not mask the skill of the uncorrected forecasts.**

**Without a proper assessment of uncorrected forecasts, it is difficult to select the appropriate downscaling model.**

See the reply to reviewer #1 on the same topic.

**2. The quality and resolution of several figures is clearly below the standard of an international scientific journal. We strongly recommend redrawing figures 1, 4 and 6, increasing the resolution and/or converting them to a vector format. Unfortunately, with such low figure quality, it becomes difficult to check the comments made by the authors in reference to those figures. Additional comments on the figures are provided in the next section to improve their readability.**

We are aware of this issue, we will improve the layout of all figures.

**3. The analysis of forecast reliability (or statistical consistency as per the author's nomenclature) lacks important information to properly assess forecast performance:**

**• It is not clear which variables are used to draw the PIT plots (figure 7). Such plots require a single series of PIT values computed from matched pairs of observations and forecasts. The authors do not precise if the observations and forecasts are coming from a single grid cell, or from a spatial aggregation (e.g. the whole Denmark). This point is important to understand their difficulties in interpreting the PIT plot (see Line 6 page 9:"issues (...) are not remarkably clear"). We suggest drawing the PIT plots for a selected set of grid points and one lead time representing the main characteristics of the PITs across the study area.**

Thanks for noticing this point. We compute the $z_i$'s (Page 6 Line 7) of N pairs of forecast-observations of each grid cell which then ends up with 662 x N values of $z_i$'s for the whole Denmark (for P, 724 for T and ET0). Then for the PIT

diagram we pool all 662 x N $z_i$'s. We are aware that by doing this we are pooling points that might not be independent which perhaps has an influence in limiting our ability to detect biases in spread (Hamil, 2000).

We then will replace Fig. 7 as well as Fig. S7 to Fig. S9 in the supplement with PIT diagrams of a selection of grid points as you suggest. We believe that this will reflect the biases in the mean observed in Fig. 4 and Fig. S1 to Fig. S3 in the supplement.

**• The analysis of reliability lacks a quantitative criterion similar to the skill scores. A standard approach is to compute the pvalue of a uniformity test such as the Kolmogorov-Smirnov test (Laio and Tamea, 2007). However, instead of the KS test, we recommend the Anderson-Darling test (Anderson and Darling, 1952), which exhibits a greater power (Noceti et al., 2003) and better ability to detect deviations from uniformity of the extremes. We strongly suggest computing the p-value of such tests for all grid points of the domain and all lead times, and then summarise the results by counting how many cells pass the test with a 5% threshold for a given lead time. This will provide a consistent and reproducible assessment of forecast reliability across the study area.**

We appreciate the suggestion and believe that will add more quantitative evidence on the discussion of statistical consistency. We will make use of the AD test for uniformity and create a figure similar to Fig. 9 for statistical consistency. The results will be then discussed in the updated version of the manuscript.

**4. Temperature is treated differently than rainfall and PET both in the computation of the bias score and in the implementation of the LS post-processing model. This is quite confusing and should be harmonised. We recommend that all bias scores be computed in relative terms to facilitate comparison. The LS model should be applied to the three variables in two modes: multiplicative (which is the configuration used for P and PET)and additive(used for T).This approach would provide a clear advice on the choice of the LS configuration.**

The configuration of the LS is as chosen because of the nature of the variables. For example, P and ET0 cannot be negative. Applying the LS as additive as you suggest can end up with negative P if the correction factor in Eq. 2. is negative (in case of underestimation) and larger than the value of the ensemble member to be corrected ($f_{k,i}$ in Eq. 2). Therefore, we have decided to keep the configuration as it is and explain the reason of the differences in implementation between variables in the updated version of the manuscript.

**5. It is not clear how extrapolation is undertaken within the QM model. The authors only indicate that "Extrapolation is then needed to map ensemble values and percentiles that are outside the training range" (see Page 4, Line 21). This is a critical and frequent problem with the QM method which affects extremes, and is then of particular interest in hydrological forecasts. Please detail the extrapolation method.**

We decided to go with a simple approach for the fitting of the empirical CDF as it has been documented that an empirical approach leads to better results (Crochemore, et al., 2016). We do recognize that this choice will have effects especially if the focus is on the evaluation of extreme values. Other approaches might be more suitable for such cases (such as fitting an extreme value distribution to extend the empirical distributions as in Wood et al., 2002). However, this is out of the scope of the present paper. We will address the limitations and advantages of our choice in the discussion of the revised manuscript.

**Considering that the paper is covering an important topic for seasonal forecasting, but that the number of points to be improved is quite significant, we recommend the paper to be accepted with major revision. Detailed comments are provided in the following section.**

**2 Specific comments**

**1. Page 2, Line 26, "The most used methods are linear scaling and quantile mapping": Zhao et al. (2017) provides an interesting perspective on the limitations of Quantile-Quantile mapping that could be cited here.**

We will cite this important paper as you suggest.

**2. Page 3, Line 4, "A statistical consistent forecast system has low (or non-existent) bias in both mean and variance": correct? ok.**

OK.

**3. Page 3, Line 14, "we make use of the Makkink equation (Hendriks, 2010) that takes as inputs temperature and incoming short-wave solar radiation from ECMWF System 4": a mention on the quality of radiation forecast would be useful. A full analysis of radiation forecast performance is out of scope here, but perhaps some references can be cited to understand the impact of radiation inputs in ET0 calculation.**

We have searched for studies with specific focus on radiation and only found the following HESS Discussion paper in this special edition.

Greuell, W., Franssen, W. H. P., Biemans, H., and Hutjes, R. W. A.: Seasonal streamflow forecasts for Europe – I. Hindcast verification with pseudo- and real observations, Hydrol. Earth Syst. Sci. Discuss., https://doi.org/10.5194/hess-2016-603, in review, 2016.

If we find more on this topic they will be cites to address the performance of radiation forecasts.

**4. Page 3, Line 20, "The time and spatial variations of the variables can be seen in Fig. 1.": Fig 1 is not readable, so the accuracy of this comment is hard to assess.**

As mentioned above, we will improve the readability of all figures presented.

**5. Page 3, Line 38, "once a scale factor has been applied": In the case of T, the factor is not a scale, but a shift.**

Thank you for noticing this. We will improve it accordingly.

**6. Page 6, Line 2, "a Wilcoxon-Mann-Whitney test was carried out.": What was the data used to apply the WMW test? Was it applied to all grid cells for a single lead time? Please clarify.**

We applied the test considering N forecast-observation pairs of each grid for a single lead time. We will clarify this in the updated version of the manuscript.

**7. Page 6, Line 39, "April shows an overestimation that might be due to the 'drizzle effect' in a month where dry days are more common.": It is quite surprising that this "drizzle effect" is not affecting the bias in March and May where the bias shows an opposite trend compared to April. This statement requires more explanations. We suggest showing the number of dry days per month, to confirm that April has the highest proportion.**

You are correct, this statement requires additional information to back up our claim. We will include this information in the updated version of the manuscript.

**8. Page 11, Line 13, "One advantage ECMWF System 4 has over ensemble climatology is that forecasts are sharper": The forecast is sharper but it is not clear if it is reliable. In this case, being sharp is not an advantage, but a problem.**

We will reformulate the sentence accordingly.

**9. Page 11, Line 24, "QM assumes that there is a linear relationship between ensemble mean and observations, assumption may not hold": We believe that the authors mean LS instead of QM here. Otherwise we do not see why QM would assume such linear relationship.**

We meant that when there is a linear relation between forecast and observations, QM performs better as has been demonstrated in Zhao et al. (2017). We will rephrase this statement accordingly.

**10. Figure5: The figure is trying to convey too many information at the same time with a confusing choice of symbol sizes (decreasing symbol size for better sharpness skill score) and color schemes. We suggest using the same presentation than Figure 3 and grouping the two figures into a single one that would provide an homogenous overview of forecast performance.**

We will separate the figures to make the readability clearer.

**11. Figure 6: The scale of the color bar is different between the 3 plots. As a result, it is impossible to compare the skill between the three variables.**

We will set the same color bar for the three variables in question.

**12. Figure 8: This plot is difficult to read because the legend does not explain that each location has a different symbol and all curves are shown with the same color. We suggest splitting each one of the three plots into 4 different subplots showing the results for one location only.**

We will state the differences more clearly in the figure caption.

**References**

Crochemore, L., Ramos, M.-H., and Pappenberger, F.: Bias correcting precipitation forecasts to improve the skill of seasonal streamflow forecasts, Hydrol. Earth Syst. Sci., 20, 3601-3618, https://doi.org/10.5194/hess-20-3601-2016, 2016.

Ferro, C. A. T., Richardson, D. S., and Weigel, A. P.: On the effect of ensemble size on the discrete and continuous ranked probability scores, Meteorol. Appl., 15, 19–24, doi:10.1002/met.45, 2008.

Hamill, T.M., 2001: Interpretation of Rank Histograms for Verifying Ensemble Forecasts. *Mon. Wea. Rev.,* **129**, 550–560, https://doi.org/10.1175/1520-0493(2001)129<0550:IORHFV>2.0.CO;2

Tian, D., C.J. Martinez, W.D. Graham, and S. Hwang, 2014: Statistical Downscaling Multimodel Forecasts for Seasonal Precipitation and Surface Temperature over the Southeastern United States. *J. Climate,* **27**, 8384–8411, https://doi.org/10.1175/JCLI-D-13-00481.1

Voisin, N., J.C. Schaake, and D.P. Lettenmaier, 2010: Calibration and Downscaling Methods for Quantitative Ensemble Precipitation Forecasts. *Wea. Forecasting,* **25**, 1603–1627, https://doi.org/10.1175/2010WAF2222367.1

Wood, A. W., E. P. Maurer, A. Kumar, and D. Lettenmaier, Long-range experimental hydrologic forecasting for the eastern United States, J. Geophys. Res., 107(D20), 4429, doi:10.1029/2001JD000659, 2002.

Zhao, T., Bennett, J. C., Wang, Q., Schepen, A., Wood, A. W., Robertson, D. E., and Ramos, M.-H. (2017). How suitable is quantile mapping for postprocessing gcm precipitation forecasts? Journal of Climate, 30(9):3185–3196.